# CoViD-19, learning from the past: A wavelet and cross-correlation analysis of the epidemic dynamics looking to emergency calls and Twitter trends in Italian Lombardy region

Bruno Alessandro Rivieccio[1]ᴼ*, Alessandra Micheletti[2]ᴼ, Manuel Maffeo[3,4], Matteo Zignani[5], Alessandro Comunian[6], Federica Nicolussi[7], Silvia Salini[7], Giancarlo Manzi[7], Francesco Auxilia[3,8], Mauro Giudici[6], Giovanni Naldi[2], Sabrina Gaito[5], Silvana Castaldi[3,9‡], Elia Biganzoli[10‡]

1 Department of Laboratory Medicine, Division of Anatomic Pathology, Niguarda Hospital, Milan, Italy, 2 Department of Environmental Science and Policy, University of Milan, Milan, Italy, 3 Department of Biomedical Sciences for Health, University of Milan, Milan, Italy, 4 Public Health Post Graduate School, University of Milan, Milan, Italy, 5 Department of Computer Science, University of Milan, Milan, Italy, 6 Department of Earth Sciences, University of Milan, Milan, Italy, 7 Department of Economics, Management and Quantitative Methods & Data Science Research Center, University of Milan, Milan, Italy, 8 ASST FBF-Sacco, Milan, Italy, 9 Fondazione IRCCS Ca' Granda Ospedale Maggiore, Milan, Italy, 10 Department of Clinical Sciences and Community Health & Data Science Research Center, University of Milan, Milan, Italy

ᴼ These authors contributed equally to this work.
‡ These authors are joint senior authors on this work. SC and EB also contributed equally to this work.
* brunoalessandro.rivieccio@ospedaleniguarda.it

**Data Availability Statement:** The data used in this study are from third party sources and can be

## Abstract

The first case of Coronavirus Disease 2019 in Italy was detected on February the 20th in Lombardy region. Since that date, Lombardy has been the most affected Italian region by the epidemic, and its healthcare system underwent a severe overload during the outbreak. From a public health point of view, therefore, it is fundamental to provide healthcare services with tools that can reveal possible new health system stress periods with a certain time anticipation, which is the main aim of the present study. Moreover, the sequence of law decrees to face the epidemic and the large amount of news generated in the population feelings of anxiety and suspicion. Considering this whole complex context, it is easily understandable how people "overcrowded" social media with messages dealing with the pandemic, and emergency numbers were overwhelmed by the calls. Thus, in order to find potential predictors of possible new health system overloads, we analysed data both from Twitter and emergency services comparing them to the daily infected time series at a regional level. Particularly, we performed a wavelet analysis in the time-frequency plane, to finely discriminate over time the anticipation capability of the considered potential predictors. In addition, a cross-correlation analysis has been performed to find a synthetic indicator of the time delay between the predictor and the infected time series. Our results show that Twitter data are more related to social and political dynamics, while the emergency calls trends can be further evaluated as a powerful tool to potentially forecast new stress periods. Since we analysed aggregated regional data, and taking into account also the huge

accessed following the protocol outlined in the Methods section and Supporting information files.

**Funding:** The authors received no specific funding for this work.

**Competing interests:** The authors have declared that no competing interests exist.

geographical heterogeneity of the epidemic spread, a future perspective would be to conduct the same analysis on a more local basis.

## Introduction

On February the 20[th] the first Italian case of Coronavirus Disease 2019 (CoViD-19) due to secondary transmission outside China was identified in Codogno, Lombardy region [1]. In the following days the number of cases started to rise not only in Lombardy but also in other Italian regions, although Lombardy remained and is still the most affected region in Italy [2]. At the time of writing (October the 4[th]), 325,329 cases have been identified in Italy, out of which 108,065 in Lombardy region [3].

The progressive decrease of CoViD-19 cases should not let our guard down, indeed it is clear that, since the beginning of Severe Acute Respiratory Syndrome Coronavirus 2 (SARS-CoV-2) pandemic, public health has suffered from the absence of a proper preparedness plan to face an episode which was unexpected and unpredictable and has heavily impacted on the territorial and hospital healthcare services. Planning has a fundamental role nowadays but it can be adequate only if the next possible pandemic peak can be effectively foreseen by means of a predictive tool which accounts for all the available signals. In order to do so, it is of paramount importance to learn from what happened during the first peak to be prepared for the potential next one.

The SARS-CoV-2 outbreak in Italy has been characterised by a massive spread of news coming from both official and unofficial sources leading to what has been defined as infodemia, an over-abundance of information—some accurate and some not—that has made hard for people to find trustworthy sources and reliable guidance needed [4].

Infodemia on SARS-CoV-2 created the perfect field to build suspicion in the population, which was scared and not prepared to face this outbreak. It is understandable how the rapid increase of the cases number, the massive spread of news and the adoption of laws to face this outbreak led to a feeling of anxiety in the population, whose everyday life changed very quickly.

A way to assess the dynamic burden of social anxiety is a context analysis of major social networks activities over the internet. To this aim Twitter represents a possible ideal tool, because of the focused role of the tweets according to the more urgent needs of information and communication, rather than general aspects of social projection and debate as in the case of Facebook, which could provide slower responses for the fast individual and social context evolution dynamics [5].

Taking into account this specific context, it is easy to understand why the 112 emergency number service in Lombardy region was suddenly overwhelmed by an enormous number of calls that rapidly overcame its capacity to cope and compromised the possibility to identify those patients who needed immediate medical assistance [6].

As pointed out by the Scientific Italian Society for Medical Emergency (SIEMS), the number of calls to 112 for the Milan province was 5,086 on February the 16[th], before the outbreak, and rapidly increased to 6,798 on February the 21[st] and to 10,657 on February the 22[nd] [7].

The emergency service in Lombardy region is organised through three first-level PSAPs (public-safety answering points), called CUR-NUE (Unique answering operating room / point —European emergency number), which forward the call to the most appropriate service, i.e. Police, Fire Department or Medical emergency rescue service. So, after the first assessment,

calls requiring medical assistance are sent to one of the four second-level PSAPs called SOREU (Regional Operating Rooms for Medical Emergency and Urgency), depending on the geographical area the call is coming from in order to evaluate the patient and decide the most appropriate intervention.

Lombardy region, from an administrative point of view, is made up of twelve provinces, and the management of medical emergency calls occur in four operating rooms which receive these calls from different sets of provinces, namely:

- SOREU delle Alpi (SRA) from the provinces of Bergamo, Brescia and Sondrio;

- SOREU dei Laghi (SRL) from the provinces of Como, Lecco and Varese;

- SOREU Metropolitana (SRM) from the provinces of Milan and Monza-Brianza;

- SOREU della Pianura (SRP) from the provinces of Cremona, Lodi, Mantova and Pavia.

Some of the non-urgent calls received by the four SOREU do not need, though, an ambulance dispatch, so not all the calls result in a medical rescue mission: in the latter cases the patients are recommended by SOREU medical technicians to consult other medical services such as general practitioners. Moreover, during the epidemic, according to specific internal procedures, SOREU medical technicians answering to the emergency calls, in the case of signs and/or symptoms evocative of CoViD-19 but not life threatening, advised patients to wait for a recall by a public health medical doctor: after this re-evaluation call, it was up to these medical doctors the final decision on the management of the case (ambulance dispatch or home quarantine).

To reduce the burden of calls to the emergency number which occurred during the first days of the outbreak, it was necessary to redirect non-urgent calls, especially those asking for information, to other services. According to European Emergency Number Association guidelines [8], Lombardy region created a regional toll-free number for CoViD-19, the first one in Italy. Other Italian regions created their own one in the following weeks, as well as other European countries like Spain, Germany, Croatia, which were facing similar issues [9].

The 24/24-hour toll-free number was settled on February the 23$^{rd}$ by AREU (Regional Emergency Service Agency) and, although it helped to funnel non-urgent calls, it was not enough because of the huge number of calls: for example on the second day it received more than 400,000 calls.

Calls to the emergency services could be an important and helpful indicator of the spread of the infection among the population, taking into account the possibility to analyse data regarding the municipality from which the calls originated and the motivations that induced people to ask for fast medical support. Statistical models could be used to assess the association of these data with new cases of CoViD-19 in order to predict new epidemic hotspots on a municipal scale, or with a smaller spatial scale for big cities.

In addition to usual public health indicators, social media data may also be used as probes of the people behaviour according to the recent trends of digital epidemiology. As mobile technology continues to evolve and proliferate, social media are expected to occupy an increasingly prominent role in the field of infectious diseases [10–12].

A recent systematic review concluded that the inclusion of online data in surveillance systems has improved the disease prediction ability over traditional syndromic surveillance systems and Twitter was the most common social network analysed for this aim [13].

Despite some limitations and concerns, a better understanding of the behavioural change induced by social media can strengthen mathematical modelling efforts and assist in the development of public policy so as to make the best use of this increasingly ubiquitous resource in controlling the spread of disease [11].

Aim of the study is to understand the correlation between the users calls to the emergency services and the spread of the infection in the population during the first peak of the CoViD-19 outbreak in Lombardy region of Italy, the first world hotspot after the Chinese raise in Wuhan. Furthermore, the joint analysis with Twitter trends related to emergency was performed to better understand the most important population concerns according to the infection dynamics. Overall, the joint active monitoring of the communication dynamics over emergency calls and social networks like Twitter could provide an integrated means for the adaptive management of information delivery as well as the optimization of the rescue logistic and finally it could provide relevant anticipation on the outbreak. These aspects appear of critical importance for CoViD-19 surveillance, and for the preparedness of emergency and strategic plans [14].

## Materials and methods

### Data

In the present work, we analysed the following time series:

- PSAP-II SOREU-118 daily incoming calls from 2020.02.18 to 2020.03.30 [15];

- PSAP-I CUR NUE-112 daily incoming calls from 2020.02.18 to 2020.03.30 [16];

- toll-free number daily incoming calls from 2020.02.23 to 2020.03.30 [16];

- daily Twitter data (tweets, replies, likes, retweets) from 2020.02.18 to 2020.06.29 [17];

- daily infected from 2020.02.24 (the first day since which Italian Department for Civil Defense has provided data) to 2020.06.29 [3].

Data about SOREU-118, NUE-112 and toll-free number daily incoming calls were only available at a regional level. The very first days of the NUE and toll-free number time series have been discarded due to the very intense population panic reaction which reflected into a very huge amount of calls (whose peak was even higher than the following, new cases-related one). Particularly, in the case of NUE they were inappropriate non-urgent calls (most of all for information need), so they were not forwarded to the corresponding SOREU: indeed, in the SOREU time series we do not observe any peak in the very first days. Moreover, this choice is justified if we consider that—in case of a new epidemic burden—there would not be such a powerful reaction, so to the aim of predictability we can take into account just the subsequent new increase in the calls to NUE and toll-free number, which is more related to the CoViD-19 dynamics. Twitter data, instead, differently from the emergency calls, were not geolocalised. Finally, daily new cases have been collected at the province level and then aggregated at the regional level.

### Twitter data analysis

The monitoring of the communication dynamics on online social media has been conducted on Twitter [18]. Specifically, the Twitter Search API (Application Programming Interface) was used to collect all the original tweets in Italian language containing the keywords "112" or "118" in the body text, discarding retweets. The data span the period from 2020.02.18 to 2020.06.29: "(112 or 118) lang:it since:2020-02-18 until:2020-06-29" was the query submitted to the Twitter Search API. We tokenised and lemmatised the text of each tweet by the POS (Part of Speech)-tagger provided by the state-of-art natural language processing (NLP) library in Python, spaCy [19]. After the lemmatisation, for each tweet we only kept the lemma of nouns and verbs. By inspecting the set of lemmas over the whole corpus of tweets, we manually

identified the most common terms related to emergency, bringing to the identification of 283 keywords (see the S1 Keywords for the list of keywords). As a rationale in the choice of the keywords, we kept terms related to: a) the reasons why people called for medical assistance (CoViD-19-related symptoms, parts of the body, words referring to family members or relatives); b) health workers or medical tools involved in the emergency calls (such as "doctor", "nurse", "oximeter"...); and c) the general context of the emergency situation (words dealing with public health policies such as "lockdown", complaints for the saturation of the emergency number services...). Finally, all the tweets that contained only the search keywords "112" or "118" in the text were excluded from the Twitter dataset, leading to a total amount of 16,216 tweets (or "statuses", in Twitter jargon) which were used for the purpose of this paper. The identification of the 283 keywords related to the emergency and the filtering based on the co-presence of these keywords with the search keywords "112" or "118" (e.g. "112" and "doctor", "118" and "ambulance"...) impacted on the reduction of the number of tweets: indeed tweets semantically not related to the topics, which however contained the terms "112" or "118", have been discarded.

Since both the timestamp and the number of likes, retweets and replies at the moment of the data collection were available for each tweet, it was possible to reconstruct four time series related to the dynamics of the emergency calls and to the perception of the situation on Twitter: 1) the production of new tweets per day, 2) the number of retweets of new tweets per day, 3) the number of likes per day, and 4) the number of replies per day. The last three indicators have been computed by aggregating the number of the interactions got by the tweets, day by day. Moreover, they capture different aspects in the perception of the emergency. Indeed, the daily number of replies is a first indicator of the level of discussions triggered by the tweets, and it might represent the reaction to a particular policy or the support of the Twitter community to other users needing medical assistance. Instead, the daily number of retweets and of likes represent an early approximation of the endorsement to the content of the tweet, and, in this specific context, a further involvement with the content of the tweets, in terms of emotional and situational closeness. For instance, a person may retweet a content reporting a call for assistance because he has experienced the same situation.

## Wavelet analysis

Wavelets represent a powerful tool to analyse localised variations of non-stationary power at many different frequencies in time series: the decomposition of the signal information in the time-frequency domain, indeed, allows to bring out variability and its changes over time [20–23]. Particularly, wavelets can "capture" and detect in the time-frequency (scale) plane both long-period and short-period trends. In other words, since the period and the frequency of a signal are inversely proportional, long-period trends correspond to low-frequency components, and are properly called "trends" or "backgrounds", while short-period trends correspond to high-frequency components, and are called "anomalies" or "discontinuities". Anomalies, despite their limited spatio-temporal location, possess a huge amount of information content, thus it is of primary importance to reveal them adequately. Among the others, the wavelet transform has relevant features such as a good capability of time-frequency localization (useful to analyse signals changing over time), and offers the possibility of a multi-resolution representation over different scales [20]. Wavelets transforms, differently from other series expansions, are thus a suitable tool to identify both a trend in a nonparametric form, keeping significant local peaks, and variations from the trend [24, 25].

Wavelet analysis is therefore useful to detect stress moments in health systems during emergencies, as in the case of CoViD-19 pandemic. By comparing peaks of two time series in a

given time interval, one can be informed with anticipation by a surrogate endpoint about the forthcoming stress peaks of the healthcare system. Namely, in the early pandemic outbreak, peaks in the emergency calls should possibly anticipate those for new infections and hospital admissions. However, for the aforementioned aims, time-frequency localization of signal components in highly discontinuous processes as a pandemic is a critical task requiring the modelling features of the wavelets analysis. Methods as those based on conventional spectral analysis such as discrete Fourier transform are not suited for the same aims since it assumes periodicity in the time series [26]. Moreover, while wavelet transform allows a variable higher time-frequency resolution according to the needs, other alternative methods which are able to detect frequencies changing over time display some limitations. Indeed, short time Fourier transform has a fixed time-frequency resolution, and Hilbert-Huang transform applied after empirical mode decomposition of the signal only computes the instantaneous frequency of each empirical mode, without highlighting the non-stationarity of the frequencies contributions [26–30].

In order to identify the trends, all the time series were first smoothed using a moving average linear filter of a 7-days amplitude, decomposed through the wavelet transforms and then normalised to their maximum values. Tweets-dependent data (daily number of replies, likes and retweets) were also previously normalised to the correspondent number of daily tweets.

Details on the adopted continuous wavelet transform (CWT) and related measures are reported in S1 File. Briefly, first of all, the CWT of all the signal was computed, together with the magnitude of the wavelet transform (using the modulus of the complex values). After that, for each pair of time series, the following quantities were calculated: the magnitude-squared wavelet coherence (MSWC), the wavelet cross-spectrum (WCS), the cone of influence, the phase coherence relationship (using the argument of the complex values), the time delay between the two signals (using the phase lag values). Each value of scale was converted to the equivalent Fourier value of frequency, and thus to the correspondent period. However, note that the relationship between scale and frequency is only an approximation, since there is not a precise correspondence between the two: among others, Meyers and coll. (1993) proposed a method for the conversion from scales to "pseudo-frequencies" [31].

Wavelet coherence and cross-spectrum analysis provides a detailed both time- and frequency-localised information on the phase lag and thus on the time delay between the compared signals, thanks to the decomposition through CWT, but lacks a global view of the trends and their relative shifts over time. Thus, in order to give a synthetic and unique indicator of the similarity over time between the signals, we also performed a time domain analysis estimating their cross-correlation sequence.

## Time lag estimation through cross-correlation analysis

In signal processing, another way to determine the similarity of two discrete time sequences is the cross-correlation. Indeed, since cross-correlation measures the relation between a vector $x$ and time-shifted (lagged) copies of another vector $y$ as a function of the lag itself, it allows also a time-delay analysis between the signals. Moreover, by revealing their relative displacement in time, the lag at which there is the maximum correlation can be considered as the time shift necessary to align the series $x$ and $y$ by sliding $y$ backward (negative lag, in case $y$ has a delay compared to $x$) or forward (positive lag, in case $y$ displays an anticipation respect to $x$). To this aim, we computed the cross-correlation function between the time series, at different lags. The lag with the maximum cross-correlation value was thus identified as the "typical time delay" leading the lagged signal.

A 90% confidence interval was computed both for the cross-correlation values and for the lag corresponding to the cross-correlation function peak, i.e. for the days of delay. We used

two different methods to calculate the interval estimation in order to compare parametric and non-parametric estimates: 1) a Fisher's *z* statistics through the Fisher's *z*-transformation [32]; 2) a Monte Carlo method through a surrogate time series with the same auto-correlation of the original one, for 1,000 simulations [33]. Details and notes on these two confidence interval computation methods are provided in S1 File.

Cross-correlation could also be estimated using wavelets, specifically through a maximal overlap discrete wavelet transform (MODWT): it would have been our intention to complete the analysis with wavelets, but since emergency calls time series are made up of few samples, there are not enough non-boundary coefficients, even at the first level, to compute the wavelet cross-correlation function for a sufficient number of lags (see explanation about the edge effects in S1 File).

All the previous reported analyses were performed using MatLab R2020a, The MathWorks Inc.

## Results

### Wavelet analysis

As already pointed out before, the following daily regional aggregated data were considered:

- toll-free number incoming calls,

- NUE-112 incoming calls,

- SOREU-118 incoming calls.

In addition, we compared also Twitter data to the number of regional daily infected patients.

The WCS and the MSWC were calculated for each of these time series in relation to the data of regional daily infected. Indeed, both wavelet cross-power spectrum and coherence, through the CWT, can show areas in the time-frequency space where two signals share common harmonic components. In particular, the focus will be on the areas for which coherence is higher than 0.5 (indicated by the arrows, see Figs 1–7), since for lower values of coherence the phase lag is not reliable.

In the following figures (Figs 1–7), the time courses of the smoothed and normalised series are displayed on the left, whereas the WCS/MSWC representation is on the right. In the time courses charts, the daily infected curve is in red, while the potential predictor curve is in blue. In the WCS/MSWC charts, the *x* axis represents time (days), the *y* axis (logarithmic scale) represents scale (which has been converted to the equivalent Fourier frequency, cycles/day), and the color scale represents the MSWC. The cone of influence, where edge effects should be considered, is shown as a white dashed line. For areas where the coherence exceeds 0.5, the charts display arrows to show the phase lag between the two signals. The arrows, which do not represent vectors since the length is not proportional to the intensity, are spaced in time and scale. The direction of the arrows designates the relative phase on the unit circle: a rightward-pointing arrow indicates in-phase coherence relationship of the two signals ($\Delta\varphi = 0$); a leftward-pointing arrow indicates anti-phase coherence relationship ($\Delta\varphi = \pi$). The corresponding lag in time depends on the duration of the cycle (period).

Looking at the time courses, it is evident a large anticipation (about two weeks) of the emergency calls trends with respect to the epidemic dynamics: if we consider the peaks of the curves, indeed, while the infected time series reaches its maximum at day 29, the toll-free number and SOREU calls peak occur at day 15 (time delay -14 days), and the NUE calls curve reaches the peak at day 16 (time delay -13 days). However, wavelet analysis does not reveal any

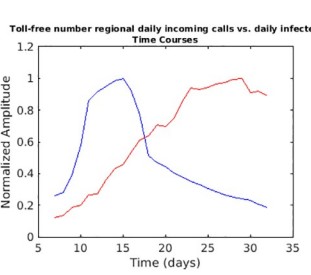
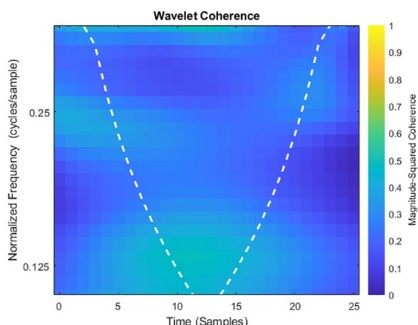

**Fig 1. Regional toll-free number daily incoming calls vs. daily infected time courses and wavelet analysis.** On the left, the smoothed (7-days amplitude moving average) and normalised time courses are displayed (toll-free number calls in blue, daily infected in red); on the right, WCS and MSWC chart is shown (see text for explanation).

strong coherence for all these pair of signals: specifically, among all the previously mentioned time series, the only one for which wavelet analysis does not display any relevant coherence compared to daily new cases data is that of regional toll-free number incoming calls (Fig 1). Instead, NUE regional data and daily infected signals (Fig 2) display coherence over days from 18 to 22 at frequencies around 0.25 cycles/day, with a phase lag from -126.4˚ to -134.7˚, corresponding to a time delay from 2.5 days to 2.6 days. Interestingly, days from 18 to 22 are confined between the two peaks, since the NUE calls curve reaches the peak at day 16, while the infected curve reaches its maximum at day 29. Not surprisingly, wavelet cross-spectrum and coherence analysis between regional daily incoming calls to SOREU and infected people (Fig 3) shows an anomaly less limited over time and over scale (frequency band ranging from 0.25 cycles/day to 0.37 cycles/day, time interval from day 17 to day 28), with a phase shift between -72.1˚ and -111.2˚, leading to a time delay of about 2.4 days.

Even if not geolocalised, we finally compared regional epidemic time series with Twitter data (Figs 4–7): just considering the time courses, it is evident that the best potential predictor is the tweets time series (peaks at day 22 and day 29, respectively for daily number of tweets and infected curve). Indeed, since replies, likes and retweets are variables dependent on the original tweets, they are delayed in time: likes and retweets reach the peak at day 28, just one day before new cases, loosing almost all the anticipation capability, while the maximum value of replies is achieved even later than the peak of the infected curve, at day 33. Wavelet analysis, focused on the predictability, confirms this consideration:

- in the case of daily tweets (Fig 4) it detects two relevant areas of high coherence in the time-frequency plane:

  - a trend at the lowest frequencies (from 0.02 cycles/day to 0.03 cycles/day), with the phase relationship shifting from in-phase coherence to a maximum lag of 31.8˚ (time delay 2.6 days),

  - and a time-localised anomaly before the infected peak around the frequency of 0.25 cycles/day, with a phase-lag ranging from 73.0˚ (day 20, 0.37 cycles/day) to 116.3˚ (day 27, 0.23 cycles/day) and a subsequent time delay between 0.5 and 1.4 days;

- for the number of daily likes (Fig 5) just a trend localised in the first half of the observation period and at the lowest frequencies is captured, with a maximum phase lag of 65.3˚ at 0.06 cycles/day, corresponding to a time shift of 3.0 days;

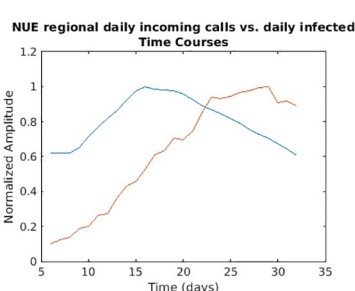
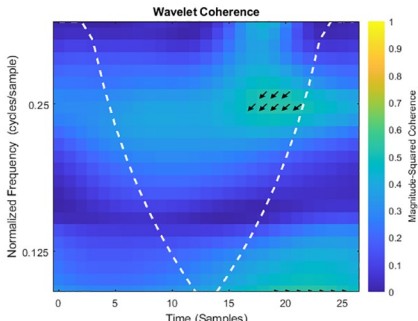

**Fig 2. Regional NUE daily incoming calls vs. daily infected time courses and wavelet analysis.** On the left, the smoothed (7-days amplitude moving average) and normalised time courses are displayed (NUE calls in blue, daily infected in red); on the right, WCS and MSWC chart is shown (see text for explanation).

- daily number of retweets (Fig 6) display a similar trend more localised in the frequencies domain (from 0.04 cycles/day to 0.08 cycles/day), with a phase shift always near 0˚ (in the range between -26.7˚ and 19.6˚);

- finally, the replies time series (Fig 7) shows a background even more localised both in time and frequency, and an anomaly around the frequency of 0.14 cycles/day from day 19 to day 33, both with a coherence value around 0.6.

## Twitter trends

The next figures (Figs 8 and 9) display raw data about Twitter trends (tweets, replies, retweets and likes).

It is evident that, concerning the dynamics of communications and the context on social media, Twitter activity (Figs 8 and 9) is not so strictly related to the epidemic dynamics, since it is triggered most of all by social, political and chronicle news, which drive an emotional participation of the users. Indeed, as highlighted by the orange box in Fig 8, the first increase in all these time series (tweets, replies, likes and retweets), from 2020/02/21 to 2020/02/25, precedes just the establishment of the red areas in Codogno and Vo' Euganeo, while the second peak of daily tweets on 2020/03/14 (red circle in Fig 8) is related to the death of an operator of SRA due to CoViD-19. Moreover, likes and retweets (Fig 9) trends look more aligned to the

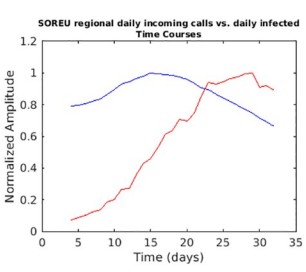
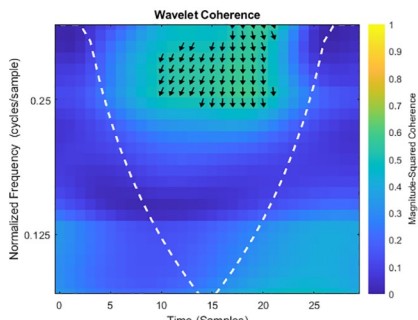

**Fig 3. Regional SOREU daily incoming calls vs. daily infected time courses and wavelet analysis.** On the left, the smoothed (7-days amplitude moving average) and normalised time courses are displayed (SOREU calls in blue, daily infected in red); on the right, WCS and MSWC chart is shown (see text for explanation).

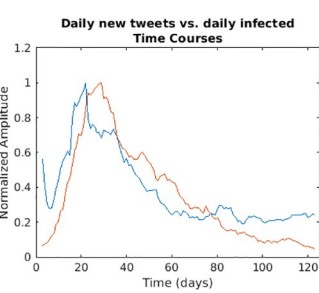
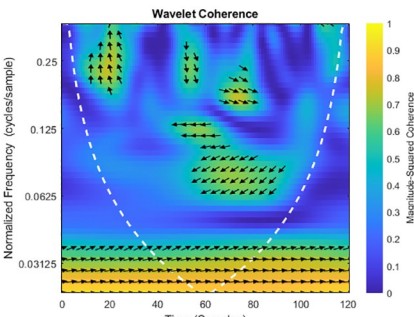

**Fig 4. Daily number of tweets vs. regional daily infected time courses and wavelet analysis.** On the left, the smoothed (7-days amplitude moving average) and normalised time courses are displayed (tweets in blue, daily infected in red); on the right, WCS and MSWC chart is shown (see text for explanation).

announcements about lockdown policies. Consequently, from the viewpoint of the management of the emergency, and comparing the Twitter trends to the emergency calls dynamics, Twitter data, despite their real-time nature, represent a weak signal for the real-time monitoring and forecasting of the outbreaks, rather they are more suitable to measure the perception of the outbreak and the lockdown policies.

## Cross-correlation and time delay analysis

While wavelet decomposition allows a detailed analysis in both the time and frequency domains, it lacks a unique, global indication of the shift over time between two signals: to this aim, we performed also a time delay analysis estimating the cross-correlation sequence for each pair of time series. This analysis has been conducted only for the three cases for which WCS/MSWC revealed a strong coherence in the ascending phase (before both the peaks or between them), which is the most important from a public health monitoring point of view. These time series are the following: (i) daily regional incoming calls to NUE-112 (Fig 10); (ii) daily regional incoming calls to SOREU-118 (Fig 11); (iii) daily number of new tweets (Fig 12). In the following figures, the maximum of each function is depicted in red, and the confidence limits for the peak lag, deduced by the 90% $z$-Fisher confidence bounds of the cross-correlation values, are reported below the figures, in the caption. Negative lags denote by how many days

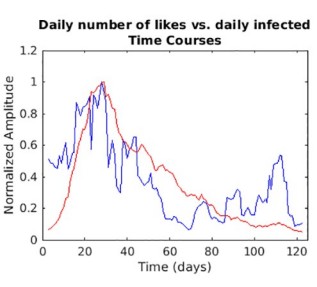
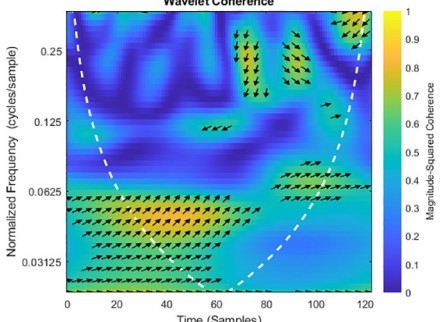

**Fig 5. Daily number of likes vs. regional daily infected time courses and wavelet analysis.** On the left, the smoothed (7-days amplitude moving average) and normalised time courses are displayed (likes in blue, daily infected in red); on the right, WCS and MSWC chart is shown (see text for explanation).

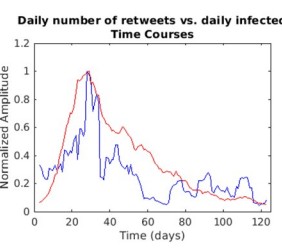
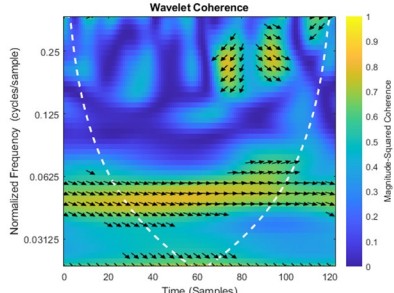

**Fig 6. Daily number of retweets vs. regional daily infected time courses and wavelet analysis.** On the left, the smoothed (7-days amplitude moving average) and normalised time courses are displayed (retweets in blue, daily infected in red); on the right, WCS and MSWC chart is shown (see text for explanation).

the time series of infected patients should be shifted backward over time to be "aligned" with the predictor.

In addition, a sensitivity analysis of these results with respect to the amplitude of the initial smoothing with a moving average filter was performed. The results are reported in Table 1.

Changes in the results when the amplitude of the window is varying are relatively small (Table 1), consequently it can be assumed that our results are robust with respect to this parameter.

Similar and consistent results have been obtained with the Monte Carlo method described in S1 File (Figs 13–15).

## Discussion

Lombardy region has been the epicentre of the CoViD-19 epidemic in the Western Countries [34]. After the detection of the first case, on February the 20th, national and regional health authorities put in place several strategies to limit the spread of the infection and deal with the consequences of the increasing number of cases [35–37]. Indicators that can reveal and anticipate a rise of cases are of paramount importance to support the planning and interventions of the health service organization. Our study shows that the number of calls to emergency services could be a good indicator that can anticipate the need for hospitalization in the early pandemic outbreak.

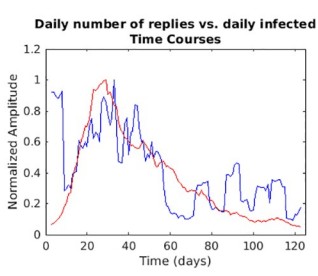
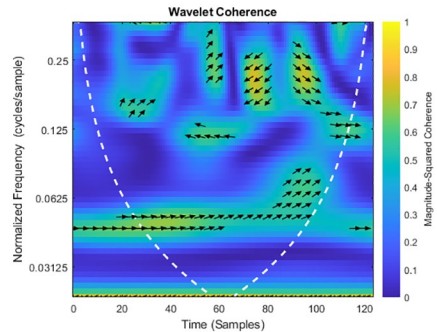

**Fig 7. Daily number of replies vs. regional daily infected time courses and wavelet analysis.** On the left, the smoothed (7-days amplitude moving average) and normalised time courses are displayed (replies in blue, daily infected in red); on the right, WCS and MSWC chart is shown (see text for explanation).

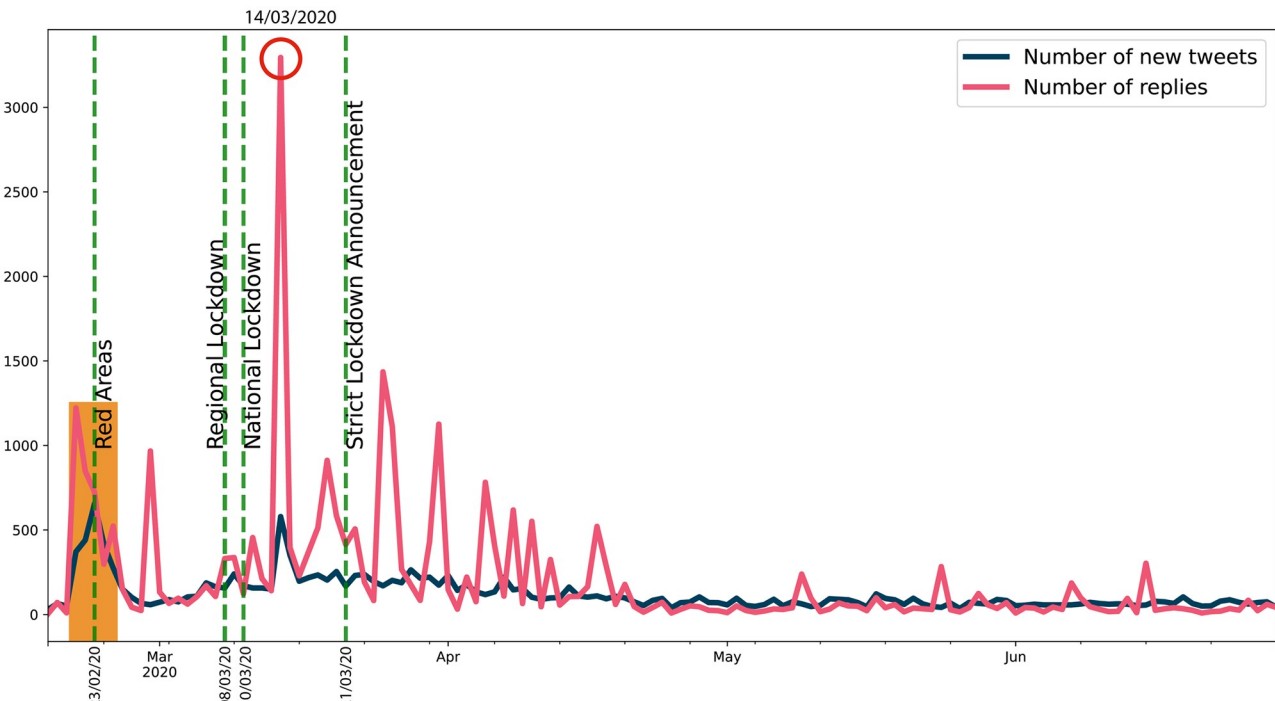

**Fig 8. Twitter trends (1).** The trends of daily number of new tweets about emergency calls (blue line) and of replies (pink line) they sparked are shown here. The vertical green dotted lines indicate the principal episodes related to the lockdown policies in Italy. The orange box indicates the first peaks of activities, while the red circle highlights the highest peak in the number of replies.

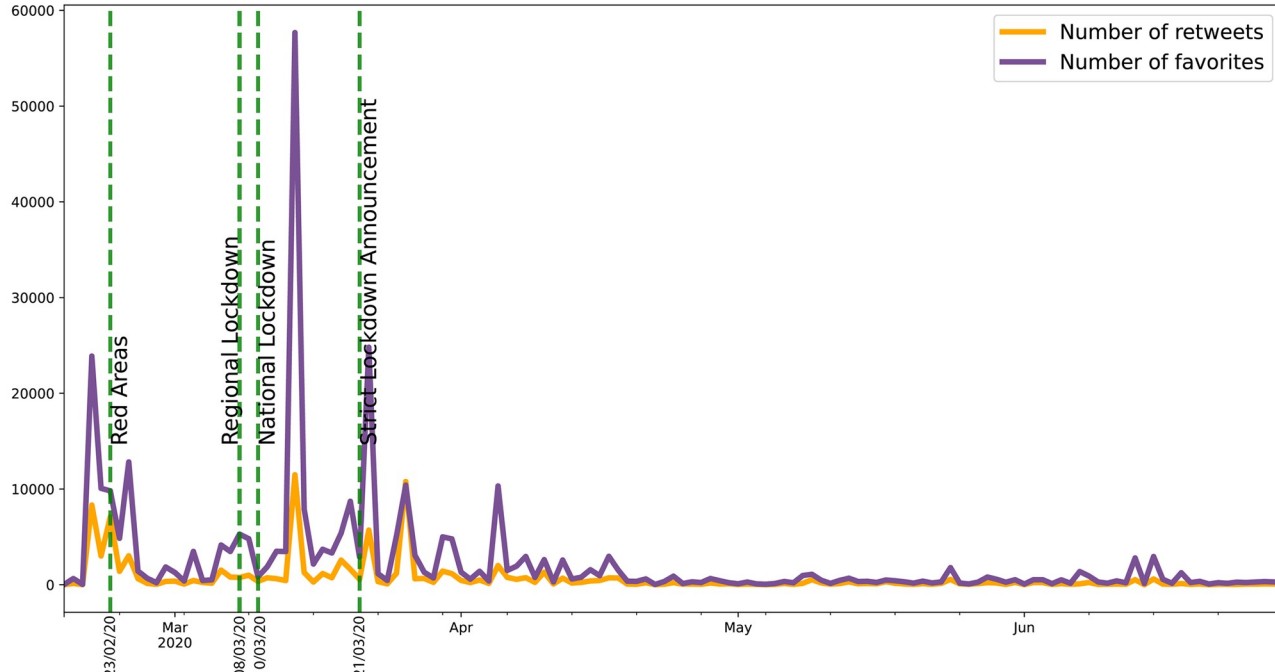

**Fig 9. Twitter trends (2).** The trends of daily number of retweets (yellow line) and of likes (purple line) are shown here.

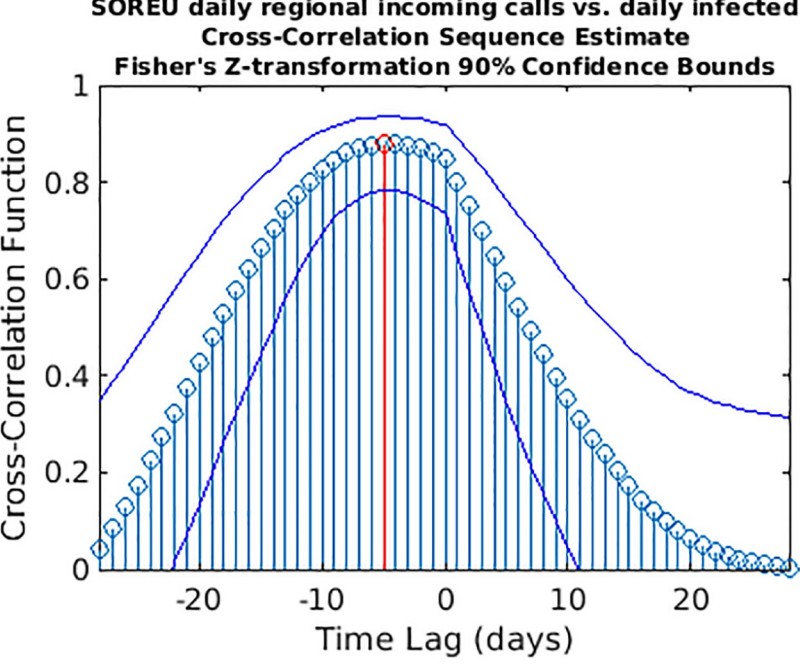

**Fig 10. Cross-correlation sequence estimate between NUE regional incoming calls and daily infected.** The blue lines represent the 90% confidence interval (CI) limits computed through a $z$-transformation. The maximum value of the cross-correlation function is depicted in red. Peak lag [CI]: -3 days [-8,1].

**Fig 11. Cross-correlation sequence estimate between SOREU regional incoming calls and daily infected.** The blue lines represent the 90% confidence interval (CI) limits computed through a $z$-transformation. The maximum value of the cross-correlation function is depicted in red. Peak lag [CI]: -5 days [-11,1].

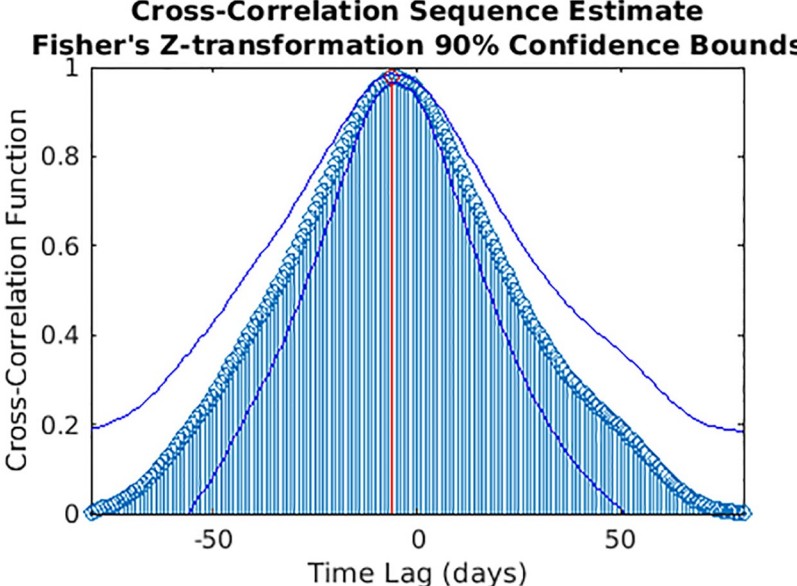

**Fig 12. Cross-correlation sequence estimate between daily number of new tweets and daily infected.** The blue lines represent the 90% confidence interval (CI) limits computed through a *z*-transformation. The maximum value of the cross-correlation function is depicted in red. Peak lag [CI]: -6 days [-8,-2].

Tô et al. [38] have recently adopted the wavelet theory to model the number $I(t)$ of infected individuals at time $t$ in the context of the classical susceptible—infected—recovered (SIR) epidemiological model, but to the best of our knowledge this is the first time that wavelet analysis is used to detect health system stressful periods during the CoViD-19 pandemic according to the coherence analysis of the peaks from different time series related to specific marker events.

However, some considerations about the analysis we conducted seem to be appropriate. With respect to wavelet analysis, wavelet decomposition in the time-scale plane has the advantage of giving a huge, precise and detailed amount of information both about time localization and frequency components. Nonetheless, this kind of analysis has also some limitations. First of all, time and frequency resolutions, according to uncertainty Heisenberg principle, are inversely proportional. Secondly, the edge effects affect the reliability of the results in the time-scale plane outside the cone of influence (see S1 File for an explanation), so that for the lowest frequencies the time interval with reliable results is very short. Lastly, one must consider also the band-pass filtering action of the CWT: indeed, the frequency spectrum bounds are a function of the number of samples of the signal (see S1 File for a detailed explanation), so the availability of data over time could represents a limitation for this kind of analysis. In our specific instance, it can be noticed that the inferior limit of the frequency domain in the case of Twitter data is much lower than the one of the daily regional emergency calls time series (112, 118, toll-free number), just because much more samples for the Twitter data are available (Figs 1–7). Consequently, in the cases of the calls to the emergency services, the ability of wavelet decomposition in revealing hidden signals and trends at the lowest frequencies is limited by the poor number of collected data. This limitation also affects the WCS/MSWC computation and the possibility of detecting large time delays between the signals. This occurs because the lowest frequencies are cut off by the wavelet band-pass filter, so the duration of a cycle corresponding to the inferior limit of the frequency spectrum (i.e. the longest period of the band) is

**Table 1. Sensitivity tests on the uncertainty in the location of the cross-correlation function peak.**

| n. days | NUE calls | | SOREU calls | | tweets | |
|---|---|---|---|---|---|---|
| | peak lag | C.I. | peak lag | C.I. | peak lag | C.I. |
| 4 | -4 | [-10,0] | -5 | [-11,1] | -6 | [-8,-3] |
| 5 | -4 | [-9,0] | -5 | [-11,1] | -6 | [-8,-3] |
| 6 | -4 | [-9,1] | -5 | [-11,1] | -5 | [-8,-2] |
| 7 | -3 | [-8,1] | -5 | [-11,1] | -6 | [-8,-2] |
| 8 | -3 | [-8,1] | -4 | [-11,1] | -5 | [-8,-2] |
| 9 | -2 | [-8,1] | -4 | [-10,0] | -5 | [-8,-2] |
| 10 | -2 | [-8,1] | -3 | [-10,1] | -5 | [-8,-2] |

Different moving-average amplitudes have been used to test the robustness of the results of the confidence intervals (C.I.) computed through the *z*-transformation.

shorter than the time delay between the two signals. Thus, the large anticipation in the ascending phase evident from the time courses (see in the results the delays between the peak days) cannot be captured by WCS/MSWC. A future perspective would be to obtain and analyse a more complete set of data over the time scale: as already explained, with a greater number of samples, indeed, the minimum spectrum frequency gets lower, the corresponding duration of a cycle (i.e. the maximum period) increases and consequently wavelet analysis through CWT could be able to detect larger time delays between the signals. Moreover, even if aggregated regional data allow us to capture the "sum" of the effects of different local situations with probable a greater anticipation capability, if we consider the enormous geographical heterogeneity of CoViD-19 spread, regional data possess a limited usefulness for public health monitoring

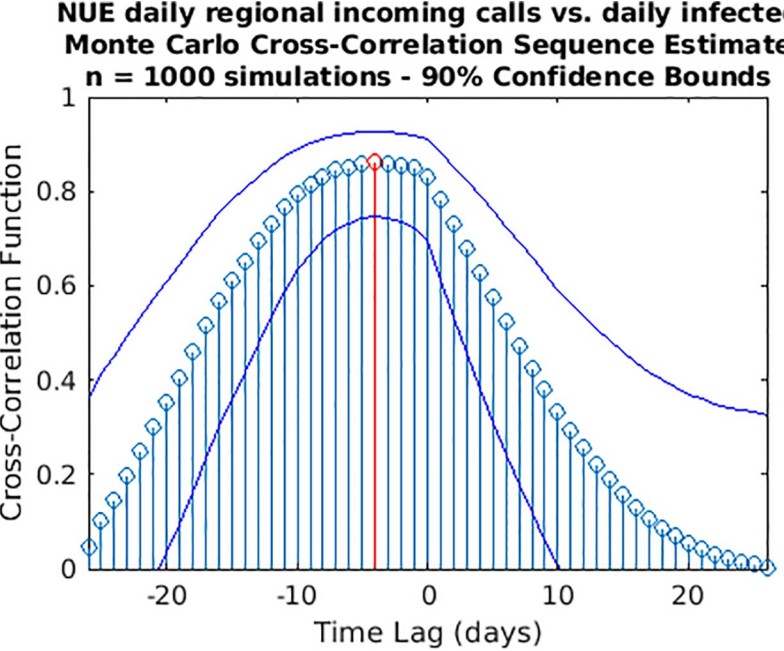

**Fig 13. Cross-correlation analysis through a Monte Carlo simulation method for the regional NUE daily incoming calls vs. the daily infected time series.** A random phase test (see S1 File) has been performed (1,000 simulations) to compute the time lag to "align" the signals and the corresponding confidence interval (C.I.): time lag = -4 days (C.I. -11,1).

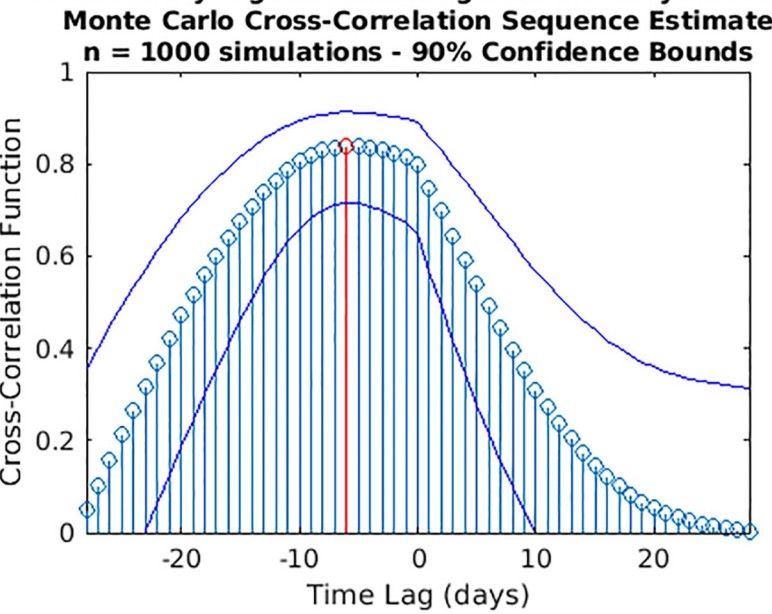

**Fig 14. Cross-correlation analysis through a Monte Carlo simulation method for the regional SOREU daily incoming calls vs. the daily infected time series.** A random phase test (see S1 File) has been performed (1,000 simulations) to compute the time lag to "align" the signals and the corresponding confidence interval (C.I.): time lag = -6 days (C.I. -13,1).

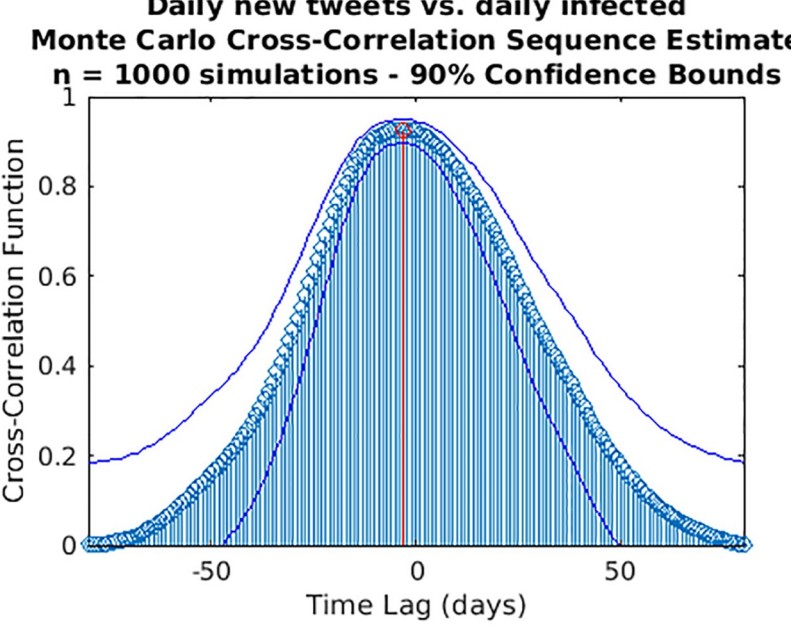

**Fig 15. Cross-correlation analysis through a Monte Carlo simulation method for the daily new tweets vs. the daily infected time series.** A random phase test (see S1 File) has been performed (1,000 simulations) to compute the time lag to "align" the signals and the corresponding confidence interval (C.I.): time lag = -3 days (C.I. -11,4).

and preparedness with respect to a possible second wave of the epidemic. One more future direction would be to analyse these data at a more local level, such as for each SOREU and each province (NUTS-3 level), or even municipalities, of Lombardy region. To supply the lack of generality of wavelet analysis, we performed also a time lag analysis through the cross-correlation function. Once again, the limited number of data affected our results: indeed, while the peaks of the emergency calls anticipated that of daily new cases by about two weeks (Figs 2 and 3), the delay of the cross-correlation function peak is much lower both for NUE calls (-3 days and -4 days, respectively for the original data and the modelled time series, see Figs 10 and 13) and for SOREU calls (-5 days and -6 days, respectively for the original data and the surrogate time series, see Figs 11 and 14). This can be explained just looking at the time courses (Figs 2 and 3): the availability of emergency calls data is such that we can observe a great part of the descending phase, whereas for the infected curve an initial decrease is visible only during the last days of the corresponding observation period. With this perspective, consequently, it would be useful to analyse a more complete dataset. Several other indicators are currently under investigation and many of them will provide useful information, but we should not only rely on indicators focused on detecting an increase in new cases, because the main impact on the health system is more related to the characteristics of the infected population rather than to the number of infected people.

The current results are referred to the early pandemic outbreak, and the same anticipation for the new cases is not necessarily expected for subsequent outbreaks since the improved capability of the healthcare systems in surveillance and early diagnosis. However, the proposed approach should be useful in forthcoming post-CoViD-19 era as a robust monitoring and convenient tool according to the needs of the early detection of new viral outbreaks. Actually, this approach has the additional advantage of a higher stability of the signal related to calls and tweets in a short term time interval: indeed, the patterns of new positive cases are extremely variable across the days of the week because of the notification delay in the reporting. On the contrary, the signal coming from the emergency calls and tweets is in real time since it is not filtered by the administrative steps and the capability of the diagnostic infrastructure.

The severe countermeasures put in place, such as the national lockdown, had a deep impact on the population from several points of view, not only on the health system. It is therefore important to take into account the social reaction to the crisis and analyzing it is part of the public health response. Our analysis shows that Twitter trends correlate more with social factors rather than with the number of cases (Figs 8 and 9). This finding suggests that a thorough analysis of social media would improve our understanding about what the most common worries, fears and feelings of the population are, in order to address them through a public health strategy that should include a proper use of social media to inform the population. Among all the Twitter data, only the daily number of new tweets reveals some anticipation capability with respect to the epidemic curve: wavelet analysis, indeed, detects a trend at the lowest frequencies, and a phase-lagged anomaly in a frequency range centred around 0.25 cycles/day that occurs just between the two peaks (Fig 4). This finding is confirmed and consistent both with the 7-days distance of the two curve peaks and with the cross-correlation analysis (maximum at -6 days and -3 days lags, respectively for the original data and the modelled time series, with small confidence intervals, see Figs 12 and 15).

## Supporting information

**S1 File. [39–42].**
(DOCX)

**S1 Data.**
(XLSX)

**S1 Keywords.**
(XLSX)

## Author Contributions

**Conceptualization:** Bruno Alessandro Rivieccio, Alessandra Micheletti, Matteo Zignani, Silvana Castaldi, Elia Biganzoli.

**Data curation:** Bruno Alessandro Rivieccio, Alessandra Micheletti, Matteo Zignani.

**Formal analysis:** Bruno Alessandro Rivieccio, Alessandra Micheletti, Matteo Zignani, Federica Nicolussi, Silvia Salini, Giancarlo Manzi, Mauro Giudici, Giovanni Naldi, Sabrina Gaito, Silvana Castaldi, Elia Biganzoli.

**Methodology:** Bruno Alessandro Rivieccio, Alessandra Micheletti, Matteo Zignani, Federica Nicolussi, Silvia Salini, Giancarlo Manzi, Mauro Giudici, Giovanni Naldi, Sabrina Gaito, Silvana Castaldi, Elia Biganzoli.

**Software:** Bruno Alessandro Rivieccio, Alessandra Micheletti, Matteo Zignani, Sabrina Gaito.

**Supervision:** Silvana Castaldi, Elia Biganzoli.

**Validation:** Federica Nicolussi, Silvia Salini, Giancarlo Manzi, Francesco Auxilia, Mauro Giudici, Giovanni Naldi, Silvana Castaldi, Elia Biganzoli.

**Writing – original draft:** Bruno Alessandro Rivieccio, Alessandra Micheletti, Manuel Maffeo, Matteo Zignani, Alessandro Comunian, Federica Nicolussi, Silvia Salini, Giancarlo Manzi, Francesco Auxilia, Mauro Giudici, Giovanni Naldi, Sabrina Gaito, Silvana Castaldi, Elia Biganzoli.

**Writing – review & editing:** Bruno Alessandro Rivieccio, Alessandra Micheletti, Manuel Maffeo, Matteo Zignani, Alessandro Comunian, Federica Nicolussi, Silvia Salini, Giancarlo Manzi, Francesco Auxilia, Mauro Giudici, Giovanni Naldi, Sabrina Gaito, Silvana Castaldi, Elia Biganzoli.

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
