## [Decision Letter · Decision Letter 0]

16 Dec 2020

PONE-D-20-32340

CoViD-19, learning from the past: a wavelet and cross-correlation analysis of the epidemic dynamics looking to emergency calls and Twitter trends in Italian Lombardy region

PLOS ONE

Dear Dr. Rivieccio,

Thank you for submitting your manuscript to PLOS ONE. After careful consideration, we feel that it has merit but does not fully meet PLOS ONE’s publication criteria as it currently stands. Therefore, we invite you to submit a revised version of the manuscript that addresses the points raised during the review process.

The paper needs a major revision in order to be evaluated for a publication. The reviewers highlighted the major issues of this work, as reported in this email.

We look forward to receiving your revised manuscript.

Kind regards,

Barbara Guidi

Academic Editor

PLOS ONE

Journal Requirements:

Reviewers' comments:

Reviewer's Responses to Questions

**Comments to the Author**

1. Is the manuscript technically sound, and do the data support the conclusions?

Reviewer #1: Yes

Reviewer #2: Partly

2. Has the statistical analysis been performed appropriately and rigorously? 

Reviewer #1: Yes

Reviewer #2: I Don't Know

3. Have the authors made all data underlying the findings in their manuscript fully available?

Reviewer #1: No

Reviewer #2: Yes

4. Is the manuscript presented in an intelligible fashion and written in standard English?

Reviewer #1: Yes

Reviewer #2: No

5. Review Comments to the Author

Reviewer #1: In this paper the authors present a study concerning the possible correlation between the spread of covid cases in Lombardy (region in Italy) and other phenomena, such as calls to the emergency numbers and twitter activity through wavelet analysis. The paper is overall clear, but I highly suggest the authors to carefully review the paper to increase the readability. In the following I highlight some weaknesses of the paper that should be addressed:

1) Page 8, line 184: how were the keywords identified? I think it's not clear from the text. Are they only keywords related to the emergency numbers? Did you try to discard the cases in which they are used with a different meaning (such as "I have 112 cats")? Is it possible to put them in a sort of table in an appendix section?

2) Page 8, lines 192-194: I am confused, are the indicators 2, 3, and 4 studied per-tweet? Or were they aggregated? Also, in both cases, how are indicators 2, 3, and 4 relevant to the scenario? The discussion is not very convincing.

3) Page 8, lines 200-202: The sentence is very long and all the parentheses make it very hard to read. Consider organise the information into two sentences: "Wavelets can [...]. In other words [...]".

4) Page 15, line 369: I notice here that you sometimes use the word "tweet", and sometimes you use "status". You should uniform the name to help readability.

5) page 15, lines 369-370: Consider using different colours, cyan and blue are similar and hinder the readability of the plots.

6) Page 16, line 380: I don't understand what are day -3 and day 1, probably you should put some of these references to the x axis of the plots. I also have questions about the findings: how much of this activity is connected to the fact that people were stuck at home "doing nothing"? And how can this analysis can help in the detection of a new outbreak (not foresee when the peak will be, but to understand that the disease is becoming a major problem)? Is it possible or not?

Reviewer #2: In this paper, authors proposes a study concerning the spread of covid cases in Lombardy (region in Italy) by analysing different data sources, such as calls to the emergency numbers and twitter activity.

The paper is not easy to follow concerning the lack of images in the right place, I suggest the authors to review the paper to improve the readability.

Furthermore, I found some points that need to be addressed:

1) Twitter Data Description: please, improve the description of the dataset, and describe how data are collected and why? In particular it is not clear why only these keywords are used. It seems to be too general to be connected with the pandemic.

2) Why wavelet are used in this analysis and how this method improves the studies concerning the pandemic. There are several works concerning COVID, please provide at least a qualitative comparison with the other methods.

3) Analysis is not clear. In particular it is not clear the efficiency of this method and how emergency calls can be used to forecast a new burden. Could be the analysis affected by the lack of specific tags?

6. PLOS authors have the option to publish the peer review history of their article (what does this mean?). If published, this will include your full peer review and any attached files.

Reviewer #1: No

Reviewer #2: No

---

## [Author Response · Author response to Decision Letter 0]

1 Feb 2021

Journal Requirements:

Our manuscript meets journal requirements.

Data are public and fully available over the internet. We have linked data sources in the References section, and we have added an Excel file named “data” as part of the Supporting Information.

Reviewers' comments:

Reviewer's Responses to Questions

Comments to the Author

1. Is the manuscript technically sound, and do the data support the conclusions?

Reviewer #1: Yes

Reviewer #2: Partly

We have added lines 421-424 (pag 17) in the Results section to stress the conclusions about Twitter data analysis:

“Consequently, from the viewpoint of the management of the emergency, and comparing the Twitter trends to the emergency calls dynamics, Twitter data, despite their real-time nature, represent a weak signal for the real-time monitoring and forecasting of the outbreaks, rather they are more suitable to measure the perception of the outbreak and the lockdown policies.”

We have specifically indicated in the Discussion that emergency calls could be a good predictor with respect to hospitalizations at least “in the early pandemic outbreak” (line 503, pag 20).

We have added lines 556-565 (pgs 22-23) in the Discussion to highlight that our findings apply to the first pandemic outbreak, explaining why the emergency calls anticipation could disappear in case of eventual future epidemic waves, the advantages of our method and the usefulness of our approach.

“The current results are referred to the early pandemic outbreak, and the same anticipation for the new cases is not necessarily expected for subsequent outbreaks since the improved capability of the healthcare systems in surveillance and early diagnosis. However, the proposed approach should be useful in forthcoming post-CoViD-19 era as a robust monitoring and convenient tool according to the needs of the early detection of new viral outbreaks. Actually, this approach has the additional advantage of a higher stability of the signal related to calls and tweets in a short term time interval: indeed, the patterns of new positive cases are extremely variable across the days of the week because of the notification delay in the reporting. On the contrary, the signal coming from the emergency calls and tweets is in real time since it is not filtered by the administrative steps and the capability of the diagnostic infrastructure.”

2. Has the statistical analysis been performed appropriately and rigorously? 

Reviewer #1: Yes

Reviewer #2: I Don't Know

Wavelet analysis has been conducted according to the international literature on this topic. 

Time-delay estimation through cross-correlation analysis has been performed computing the uncertainty range in two ways:

1. the Fisher’s z-transformation (which is the “gold standard” for cross-correlation, according to literature);

2. a Monte Carlo simulation via surrogate time-series derived through a method specific for serially correlated data (“Fourier transform random phase test”, see Supporting Information and reference #32).

We have also conducted a sensitivity test on the uncertainty in the location of the cross-correlation function peak by changing the moving average filter amplitude (see Table 1, pag 19).

3. Have the authors made all data underlying the findings in their manuscript fully available?

Reviewer #1: No

Reviewer #2: Yes

Data are public and fully available over the internet. We have linked data sources in the References section (see references #3, #15, #16, #17), and we have added an Excel file named “data” as part of the Supporting Information.

4. Is the manuscript presented in an intelligible fashion and written in standard English?

Reviewer #1: Yes

Reviewer #2: No

We have revised the manuscript in order to spot residual errors.

5. Review Comments to the Author

Reviewer #1: In this paper the authors present a study concerning the possible correlation between the spread of covid cases in Lombardy (region in Italy) and other phenomena, such as calls to the emergency numbers and twitter activity through wavelet analysis. The paper is overall clear, but I highly suggest the authors to carefully review the paper to increase the readability. In the following I highlight some weaknesses of the paper that should be addressed:

1) Page 8, line 184: how were the keywords identified? I think it's not clear from the text. Are they only keywords related to the emergency numbers? Did you try to discard the cases in which they are used with a different meaning (such as "I have 112 cats")? Is it possible to put them in a sort of table in an appendix section?

The subsection “Twitter data analysis” (Materials and methods) has been fully revised. Particularly, we have detailed the pipeline for the extraction of the lemmas from each tweet and the rationale in the choice of the nouns and verbs related to the issue, leading to a total amount of 283 keywords. We have also explained how we filtered the tweets: we selected those ones in which the search keywords “112” or “118” were co-present with one of the aforementioned 283 keywords (e.g. “112” and “doctor”, “118” and “ambulance”…). Finally, we have specified that in this way we have reduced the number of tweets, discarding those ones whose text was not related to the issue faced in our work because the terms were used with a different meaning.

Thus, lines 179-201 (pag 8) have been modified as it follows:

“The monitoring of the communication dynamics on online social media has been conducted on Twitter [18]. Specifically, the Twitter Search API (Application Programming Interface) was used to collect all the original tweets in Italian language containing the keywords “112” or “118” in the body text, discarding retweets. The data span the period from 2020.02.18 to 2020.06.29: “(112 or 118) lang:it since:2020-02-18 until:2020-06-29” was the query submitted to the Twitter Search API. We tokenized and lemmatized the text of each tweet by the POS (Part of Speech)-tagger provided by the state-of-art natural language processing (NLP) library in Python, spaCy [19]. After the lemmatization, for each tweet we only kept the lemma of nouns and verbs. By inspecting the set of lemmas over the whole corpus of tweets, we manually identified the most common terms related to emergency, bringing to the identification of 283 keywords (see the Support Information for the list of keywords). As a rationale in the choice of the keywords, we kept terms related to: a) the reasons why people called for medical assistance (CoViD-19-related symptoms, parts of the body, words referring to family members or relatives); b) health workers or medical tools involved in the emergency calls (such as “doctor”, “nurse”, “oximeter”...); and c) the general context of the emergency situation (words dealing with public health policies such as “lockdown”, complaints for the saturation of the emergency number services...). Finally, all the tweets that contained only the search keywords “112” or “118” in the text were excluded from the Twitter dataset, leading to a total amount of 16,216 tweets (or “statuses”, in Twitter jargon) which were used for the purpose of this paper. The identification of the 283 keywords related to the emergency and the filtering based on the co-presence of these keywords with the search keywords “112” or “118” (e.g. “112” and “doctor”, “118” and “ambulance”...) impacted on the reduction of the number of tweets: indeed tweets semantically not related to the topics, which however contained the terms “112” or “118”, have been discarded.”

We have also added the list of keywords as a separated Excel file named “keyword” in the Support Information.

2) Page 8, lines 192-194: I am confused, are the indicators 2, 3, and 4 studied per-tweet? Or were they aggregated? Also, in both cases, how are indicators 2, 3, and 4 relevant to the scenario? The discussion is not very convincing.

In the revised version of the paper, we have specified that all the data about tweets interactions have been aggregated, rather than considered per tweet. We have also highlighted the relevance of the indicators 2, 3, and 4 to the scenario in the subsection “Twitter data analysis” (Materials and methods). In general, these indicators may capture the perception of the issue by that part of the Twitter community which produces a few contents but shares an emotional and situational closeness to the tweets produced by other members.

Thus, lines 202-214 (pag 9) have been modified as it follows:

“Since both the timestamp and the number of likes, retweets and replies at the moment of the data collection were available for each tweet, it was possible to reconstruct four time series related to the dynamics of the emergency calls and to the perception of the situation on Twitter: 1) the production of new tweets per day, 2) the number of retweets of new tweets per day, 3) the number of likes per day, and 4) the number of replies per day. The last three indicators have been computed by aggregating the number of the interactions got by the tweets, day by day. Moreover, they capture different aspects in the perception of the emergency. Indeed, the daily number of replies is a first indicator of the level of discussions triggered by the tweets, and it might represent the reaction to a particular policy or the support of the Twitter community to other users needing medical assistance. Instead, the daily number of retweets and of likes represent an early approximation of the endorsement to the content of the tweet, and, in this specific context, a further involvement with the content of the tweets, in terms of emotional and situational closeness. For instance, a person may retweet a content reporting a call for assistance because he has experienced the same situation.”

3) Page 8, lines 200-202: The sentence is very long and all the parentheses make it very hard to read. Consider organise the information into two sentences: "Wavelets can [...]. In other words [...]".

As suggested by the reviewer, the information has been “split” into two separate sentences, as it follows (lines 220-224, pag 9):

“Particularly, wavelets can “capture” and detect in the time-frequency (scale) plane both long-period and short-period trends. In other words, since the period and the frequency of a signal are inversely proportional, long-period trends correspond to low-frequency components, and are properly called “trends” or “backgrounds”, while short-period trends correspond to high-frequency components, and are called “anomalies” or “discontinuities”.”

4) Page 15, line 369: I notice here that you sometimes use the word "tweet", and sometimes you use "status". You should uniform the name to help readability.

In compliance with the reviewer’s suggestion, we have uniformed the name as “tweet(s)”, referring to it also as “statuses” just in the line 196 (pag 8) to clarify they are synonyms in Twitter jargon.

5) page 15, lines 369-370: Consider using different colours, cyan and blue are similar and hinder the readability of the plots.

We have changed the colours of Figs 8-9: in Fig 8 cyan has been changed to blue, and blue has been changed to pink; in Fig 9 green has been changed to yellow, and red has been changed to purple. Moreover, we have inserted some important dates and highlighted the periods of high activities we have commented in the Results section.

The corresponding captions of Figs 8-9 have been modified as it follows (lines 405-411, pag 17):

“Fig 8. Twitter trends (1). The trends of daily number of new tweets about emergency calls (blue line) and of replies (pink line) they sparked are shown here. The vertical green dotted lines indicate the principal episodes related to the lockdown policies in Italy. The orange box indicates the first peaks of activities, while the red circle highlights the highest peak in the number of replies.

Fig 9. Twitter trends (2). The trends of daily number of retweets (yellow line) and of likes (purple line) are shown here.”

6) Page 16, line 380: I don't understand what are day -3 and day 1, probably you should put some of these references to the x axis of the plots. I also have questions about the findings: how much of this activity is connected to the fact that people were stuck at home "doing nothing"? And how can this analysis can help in the detection of a new outbreak (not foresee when the peak will be, but to understand that the disease is becoming a major problem)? Is it possible or not?

We have modified the plot according to the reviewer’s suggestions. Specifically, we have displayed the dates and highlighted the periods related to the highest peaks. As described in the previous response, we have also modified the caption of Fig 8. In the text we have substituted “day -3”, “day 1” and “day 20” with the dates of the events. Finally, we have added a short sentence to discuss the limitations of Twitter data and the appropriate contexts for their usage as a suitable signal.

In the end, lines 413-424 (pag 17) have been modified as it follows:

“It is evident that, concerning the dynamics of communications and the context on social media, Twitter activity (Figs 8–9) is not so strictly related to the epidemic dynamics, since it is triggered most of all by social, political and chronicle news, which drive an emotional participation of the users. Indeed, as highlighted by the orange box in Fig 8, the first increase in all these time series (tweets, replies, likes and retweets), from 2020/02/21 to 2020/02/25, precedes just the establishment of the red areas in Codogno and Vo’ Euganeo, while the second peak of daily tweets on 2020/03/14 (red circle in Fig 8) is related to the death of an operator of SRA due to CoViD-19. Moreover, likes and retweets (Fig 9) trends look more aligned to the announcements about lockdown policies. Consequently, from the viewpoint of the management of the emergency, and comparing the Twitter trends to the emergency calls dynamics, Twitter data, despite their real-time nature, represent a weak signal for the real-time monitoring and forecasting of the outbreaks, rather they are more suitable to measure the perception of the outbreak and the lockdown policies.”

Reviewer #2: In this paper, authors proposes a study concerning the spread of covid cases in Lombardy (region in Italy) by analysing different data sources, such as calls to the emergency numbers and twitter activity.

The paper is not easy to follow concerning the lack of images in the right place, I suggest the authors to review the paper to improve the readability.

We have uploaded the images as separate files and put the corresponding captions in the right place in the text, according to the Journal Requirements.

Furthermore, I found some points that need to be addressed:

1) Twitter Data Description: please, improve the description of the dataset, and describe how data are collected and why? In particular it is not clear why only these keywords are used. It seems to be too general to be connected with the pandemic.

The subsection “Twitter data analysis” (Materials and methods) has been fully revised. Particularly, we have detailed the gathering process by specifying the query submitted to the Twitter Search API and the text processing pipeline for the extraction of the lemmas from each tweet. Moreover, we have clarified the rationale in the choice of the 283 keywords related to the emergency calls. We have also explained how we filtered the tweets: we selected those ones in which the search keywords “112” or “118” were co-present with one of the aforementioned 283 keywords (e.g. “112” and “doctor”, “118” and “ambulance”…). Finally, we have specified that in this way we have reduced the number of tweets, discarding those ones whose text was not related to the issue faced in our work because the terms were used with a different meaning.

Thus, lines 179-201 (pag 8) have been modified as it follows:

“The monitoring of the communication dynamics on online social media has been conducted on Twitter [18]. Specifically, the Twitter Search API (Application Programming Interface) was used to collect all the original tweets in Italian language containing the keywords “112” or “118” in the body text, discarding retweets. The data span the period from 2020.02.18 to 2020.06.29: “(112 or 118) lang:it since:2020-02-18 until:2020-06-29” was the query submitted to the Twitter Search API. We tokenized and lemmatized the text of each tweet by the POS (Part of Speech)-tagger provided by the state-of-art natural language processing (NLP) library in Python, spaCy [19]. After the lemmatization, for each tweet we only kept the lemma of nouns and verbs. By inspecting the set of lemmas over the whole corpus of tweets, we manually identified the most common terms related to emergency, bringing to the identification of 283 keywords (see the Support Information for the list of keywords). As a rationale in the choice of the keywords, we kept terms related to: a) the reasons why people called for medical assistance (CoViD-19-related symptoms, parts of the body, words referring to family members or relatives); b) health workers or medical tools involved in the emergency calls (such as “doctor”, “nurse”, “oximeter”...); and c) the general context of the emergency situation (words dealing with public health policies such as “lockdown”, complaints for the saturation of the emergency number services...). Finally, all the tweets that contained only the search keywords “112” or “118” in the text were excluded from the Twitter dataset, leading to a total amount of 16,216 tweets (or “statuses”, in Twitter jargon) which were used for the purpose of this paper. The identification of the 283 keywords related to the emergency and the filtering based on the co-presence of these keywords with the search keywords “112” or “118” (e.g. “112” and “doctor”, “118” and “ambulance”...) impacted on the reduction of the number of tweets: indeed tweets semantically not related to the topics, which however contained the terms “112” or “118”, have been discarded.”

2) Why wavelet are used in this analysis and how this method improves the studies concerning the pandemic. There are several works concerning COVID, please provide at least a qualitative comparison with the other methods.

We have explained why wavelet analysis is useful in this context, highlighting how signal decomposition through wavelet transform can help in detecting health systems periods of stress. As suggested by the reviewer, we have also provided a qualitative comparison between wavelet decomposition and other time-frequency series expansions methods, such as discrete Fourier transform, short-time Fourier transform, and empirical mode decomposition plus Hilbert-Huang transform.

Thus, lines 231-245 (pag 10) have been added to the “Wavelet analysis” subsection (Materials and methods):

“Wavelet analysis is therefore useful to detect stress moments in health systems during emergencies, as in the case of CoViD-19 pandemic. By comparing peaks of two time series in a given time interval, one can be informed with anticipation by a surrogate endpoint about the forthcoming stress peaks of the healthcare system. Namely, in the early pandemic outbreak, peaks in the emergency calls should possibly anticipate those for new infections and hospital admissions. However, for the aforementioned aims, time-frequency localization of signal components in highly discontinuous processes as a pandemic is a critical task requiring the modelling features of the wavelets analysis. Methods as those based on conventional spectral analysis such as discrete Fourier transform are not suited for the same aims since it assumes periodicity in the time series [26]. Moreover, while wavelet transform allows a variable higher time-frequency resolution according to the needs, other alternative methods which are able to detect frequencies changing over time display some limitations. Indeed, short time Fourier transform has a fixed time-frequency resolution, and Hilbert-Huang transform applied after empirical mode decomposition of the signal only computes the instantaneous frequency of each empirical mode, without highlighting the non-stationarity of the frequencies contributions [26-30].”

Finally, we have added a short paragraph in the Discussion section in which we have cited the only paper dealing with CoViD-19 whose authors have used wavelet to construct a SIR model of the epidemic. In this paragraph we have highlighted it is the first time, to the best of our knowledge, that wavelet analysis is used to possibly detect stress moments of health services. 

Thus, lines 504-508 (pgs 20-21) have been added:

“Tô et al. [38] have recently adopted the wavelet theory to model the number I(t) of infected individuals at time t in the context of the classical susceptible – infected – recovered (SIR) epidemiological model, but to the best of our knowledge this is the first time that wavelet analysis is used to detect health system stressful periods during the CoViD-19 pandemic according to the coherence analysis of the peaks from different time series related to specific marker events.”

3) Analysis is not clear. In particular it is not clear the efficiency of this method and how emergency calls can be used to forecast a new burden. Could be the analysis affected by the lack of specific tags?

As explained in the previous response, we have clarified how this method could be used to forecast the epidemic dynamics, particularly the need for hospitalization and the subsequent potential health system stress, at least “in the early pandemic outbreak” (line 502, pag 20). In particular, we have highlighted this issue in lines 231-235 (“Wavelet analysis” subsection of Materials and methods, pag 10):

“Wavelet analysis is therefore useful to detect stress moments in health systems during emergencies, as in the case of CoViD-19 pandemic. By comparing peaks of two time series in a given time interval, one can be informed with anticipation by a surrogate endpoint about the forthcoming stress peaks of the healthcare system. Namely, in the early pandemic outbreak, peaks in the emergency calls should possibly anticipate those for new infections and hospital admissions.”

We have also added lines 556-565 (pgs 22-23) in the Discussion to highlight that our findings apply to the first pandemic outbreak, explaining why the emergency calls anticipation could disappear in case of eventual future epidemic waves, the advantages of our method and the usefulness of our approach.

“The current results are referred to the early pandemic outbreak, and the same anticipation for the new cases is not necessarily expected for subsequent outbreaks since the improved capability of the healthcare systems in surveillance and early diagnosis. However, the proposed approach should be useful in forthcoming post-CoViD-19 era as a robust monitoring and convenient tool according to the needs of the early detection of new viral outbreaks. Actually, this approach has the additional advantage of a higher stability of the signal related to calls and tweets in a short term time interval: indeed, the patterns of new positive cases are extremely variable across the days of the week because of the notification delay in the reporting. On the contrary, the signal coming from the emergency calls and tweets is in real time since it is not filtered by the administrative steps and the capability of the diagnostic infrastructure.”

Moreover, with respect to Twitter data, we have discussed the results about the identification of the peaks in Twitter trends and the lack of meaningful correlation with the emergency call time-series. We have stressed the limitation of the Twitter data for forecasting the outbreaks and highlighted a proper usage of the Twitter to understand the perception of the outbreak and of the lockdown policies by the population. 

In the Results section we have added the following sentence (lines 421-424, pag 17):

“Consequently, from the viewpoint of the management of the emergency, and comparing the Twitter trends to the emergency calls dynamics, Twitter data, despite their real-time nature, represent a weak signal for the real-time monitoring and forecasting of the outbreaks, rather they are more suitable to measure the perception of the outbreak and the lockdown policies.”

Finally, in the text analysis we have not relied on hashtags, rather we have tagged each tweet with a set of specific keywords, because the usage of hashtags is user-dependent, therefore not all the relevant keywords may be captured. The use of keywords-specific tags has allowed us an approach less related to the Twitter users’ behaviour.

6. PLOS authors have the option to publish the peer review history of their article (what does this mean?). If published, this will include your full peer review and any attached files.

Do you want your identity to be public for this peer review? For information about this choice, including consent withdrawal, please see our Privacy Policy.

Reviewer #1: No

Reviewer #2: No

We would like the full peer review history of our paper to be published, in order to increase transparency and accountability of our scientific research, reinforcing its validity.

---

## [Decision Letter · Decision Letter 1]

15 Feb 2021

CoViD-19, learning from the past: a wavelet and cross-correlation analysis of the epidemic dynamics looking to emergency calls and Twitter trends in Italian Lombardy region

PONE-D-20-32340R1

Dear Dr. Rivieccio,

We’re pleased to inform you that your manuscript has been judged scientifically suitable for publication and will be formally accepted for publication once it meets all outstanding technical requirements.

Kind regards,

Barbara Guidi

Academic Editor

PLOS ONE

Additional Editor Comments (optional):

Reviewers' comments:

Reviewer's Responses to Questions

**Comments to the Author**

1. If the authors have adequately addressed your comments raised in a previous round of review and you feel that this manuscript is now acceptable for publication, you may indicate that here to bypass the “Comments to the Author” section, enter your conflict of interest statement in the “Confidential to Editor” section, and submit your "Accept" recommendation.

Reviewer #1: All comments have been addressed

Reviewer #2: All comments have been addressed

2. Is the manuscript technically sound, and do the data support the conclusions?

Reviewer #1: Yes

Reviewer #2: Yes

3. Has the statistical analysis been performed appropriately and rigorously? 

Reviewer #1: Yes

Reviewer #2: Yes

4. Have the authors made all data underlying the findings in their manuscript fully available?

Reviewer #1: Yes

Reviewer #2: Yes

5. Is the manuscript presented in an intelligible fashion and written in standard English?

Reviewer #1: Yes

Reviewer #2: Yes

6. Review Comments to the Author

Reviewer #1: The paper is now mature enough to be accepted for publication.

I encourage the authors to perform a final round of proof reading to correct the last errors. Here are some I found (mostly connected to the usage of British English):

p 5, l 124: "to analyze data" -> "to analyse data"

p 8, l 187: "the lemmatization, for" -> "the lemmatisation, for"

p 9, l 222: "to analyze localised" -> "to analyse localised"

p 10, l 234: "analyze signals" -> "analyse signals"

p 22, l 537: "and analyze a" -> "and analyse a"

p 22, l 545: "to analyze these" -> "to analyse these"

p 23, l 557: "to analyze a" -> to analyse a"

Reviewer #2: All the previous comments have been addressed, and the contribution of the paper has been increased. For this reason the paper could be accepted as it.

7. PLOS authors have the option to publish the peer review history of their article (what does this mean?). If published, this will include your full peer review and any attached files.

Reviewer #1: No

Reviewer #2: No

---

## [Editor Report · Acceptance letter]

18 Feb 2021

PONE-D-20-32340R1 

CoViD-19, learning from the past: A wavelet and cross-correlation analysis of the epidemic dynamics looking to emergency calls and Twitter trends in Italian Lombardy region 

Dear Dr. Rivieccio:

I'm pleased to inform you that your manuscript has been deemed suitable for publication in PLOS ONE. Congratulations! Your manuscript is now with our production department. 

Kind regards, 

on behalf of

Dr. Barbara Guidi 

Academic Editor

PLOS ONE